# Computationally Selected Multivalent HIV-1 Subtype C Vaccine Protects Against Heterologous SHIV Challenge

**DOI:** 10.3390/vaccines13030231

**Published:** 2025-02-24

**Authors:** Dieter Mielke, Marina Tuyishime, Natasha S. Kelkar, Yunfei Wang, Robert Parks, Sampa Santra, Wes Rountree, LaTonya D. Williams, Tiffany Peters, Nathan Eisel, Sheetal Sawant, Lu Zhang, Derrick Goodman, Shalini Jha, Adam Zalaquett, Pratamesh Ramasubramanian, Sherry Stanfield-Oakley, Gary Matyas, Zoltan Beck, Mangala Rao, Julie Ake, Thomas N. Denny, David C. Montefiori, Margaret E. Ackerman, Lawrence Corey, Georgia D. Tomaras, Bette T. Korber, Barton F. Haynes, Xiaoying Shen, Guido Ferrari

**Affiliations:** 1Duke Human Vaccine Institute, Duke University, Durham, NC 27710, USA; dieter.mielke@duke.edu (D.M.); marina.tuyishime@duke.edu (M.T.); yunfei.wang@duke.edu (Y.W.); rob.parks@duke.edu (R.P.); wes.rountree@duke.edu (W.R.); latonya.williams@duke.edu (L.D.W.); tdpeters@email.unc.edu (T.P.); neisel@carolinas.vcom.edu (N.E.); sheetal.sawant@duke.edu (S.S.); lu.zhang809@duke.edu (L.Z.); derrick.goodman@duke.edu (D.G.); thomas.denny@duke.edu (T.N.D.); david.montefiori@duke.edu (D.C.M.); gdt@duke.edu (G.D.T.); barton.haynes@duke.edu (B.F.H.); 2Duke Center for Human Systems Immunology, Duke University, Durham, NC 27701, USA; 3Department of Surgery, Duke University, Durham, NC 27710, USA; shalini.jha@duke.edu (S.J.); adam.zalaquett@duke.edu (A.Z.); pratamesh.ramasubramanian@gmail.com (P.R.); sherry.oakley@duke.edu (S.S.-O.); 4Department of Microbiology and Immunology, Geisel School of Medicine at Dartmouth, Dartmouth College, Hanover, NH 03756, USA; natasha.s.kelkar.gr@dartmouth.edu (N.S.K.); margaret.e.ackerman@dartmouth.edu (M.E.A.); 5Beth Israel Deaconess Medical Center, Harvard Medical School, Boston, MA 02215, USA; ssantra@bidmc.harvard.edu; 6U.S. Military HIV Research Program, Walter Reed Army Institute of Research, Silver Spring, MD 20910, USA; gmatyas@hivresearch.org (G.M.); zoltan.beck@gmail.com (Z.B.); mrao@hivresearch.org (M.R.); jake@hivresearch.org (J.A.); 7Thayer School of Engineering, Dartmouth College, Hanover, NH 03755, USA; 8Vaccine and Infectious Disease Division, Fred Hutchinson Cancer Center, Seattle, WA 98104, USA; lcorey@fredhutch.org; 9Department of Integrative Immunobiology, Duke University School of Medicine, Durham, NC 27710, USA; 10Department of Molecular Genetics and Microbiology, Duke University School of Medicine, Durham, NC 27710, USA; 11T-6: Theoretical Biology and Biophysics, Los Alamos National Laboratory, Los Alamos, NM 87545, USA; btk@lanl.gov; 12New Mexico Consortium, Los Alamos, NM 87545, USA

**Keywords:** HIV-1, subtype C, variable loops 1 and 2, correlate of protection, systems serology

## Abstract

**Background**: The RV144 trial in Thailand is the only HIV-1 vaccine efficacy trial to date to demonstrate any efficacy. Genetic signatures suggested that antibodies targeting the variable loop 2 (V2) of the HIV-1 envelope played an important protective role. The ALVAC prime and protein boost follow-up trial in southern Africa (HVTN702) failed to show any efficacy. One hypothesis for this is the greater diversity of subtype C viruses in southern Africa relative to CRF01_AE in Thailand. **Methods**: Here, we determined whether an ALVAC prime with computationally selected gp120 boost immunogens maximizing coverage of diversity of subtype C viruses in the variable V1 and V2 regions (V1V2) improved the protection of non-human primates (NHPs) from a heterologous subtype C SHIV challenge compared to more traditional regimens. **Results**: An ALVAC prime with Trivalent subtype C gp120 boosts resulted in statistically significant protection from repeated intrarectal SHIV challenges compared to the control. Evaluation of the immunogenicity of each vaccine regimen at the time of challenge demonstrated that different gp120 combination boosts elicited similar high magnitudes of gp120 and breadth of V1V2-binding antibodies, as well as strong Fc-mediated immune responses. Low-to-no neutralization of the challenge virus was detected. A Cox proportional hazard analysis of five pre-selected immune parameters at the time of challenge identified ADCC against the challenge envelope as a correlate of protection. Systems serology analysis revealed that immune responses elicited by the different vaccine regimens were distinct and identified further correlates of resistance to infection. **Conclusions**: Computationally designed vaccines with maximized subtype C V1V2 coverage mediated protection of NHPs from a heterologous Tier-2 subtype C SHIV challenge.

## 1. Introduction

The RV144 vaccine efficacy trial tested a canarypox-protein HIV vaccine regimen (ALVAC-HIV and AIDSVAX B/E gp120) and is the only HIV trial to date to demonstrate partial vaccine efficacy, with a modest 31.2% estimated protection at 42 months [1]. An initial analysis showed that IgG responses binding to the variable loops 1 and 2 (V1V2) and antibody-dependent cellular cytotoxicity (ADCC) responses in the presence of low anti-envelope IgA responses were correlated with reduced risk of infection [2]. A follow-up study also indicated that decreased risk of infection was associated with higher response rates and magnitudes of IgG3 V1V2-specific antibodies [3].

HIV remains a significant health burden in southern Africa. In 2023, southern and eastern Africa accounted for 35% (450,000) and 41% (260,000) of new infections and HIV-related deaths, respectively [4]. Consequently, there is a significant focus on a protective vaccine specific for this region. In contrast to Thailand, where the circulating recombinant strain CRF01_AE is most prevalent, southern Africa has very high levels of HIV-1 subtype C. Accordingly, several vaccine concepts have been developed using subtype C immunogens. HVTN702 (NTC02968849) was a phase-IIb/III trial designed to test a subtype C specific regimen analogous to the RV144 regimen (ALVAC-C prime with an MF59-adjuvanted Bivalent subtype C gp120 boost) but did not show any efficacy and so was stopped early [5]. Similarly, the phase-IIb Imbokodo vaccine efficacy trial (HVTN705; NTC03060629), which consisted of an adenovirus-26 mosaic HIV vector and a subtype C gp140 boost, did not show any efficacy [6]. Both vaccines elicited lower magnitude of V1V2 responses than RV144 but, in separate correlates analyses, this response was still associated with decreased risk of HIV-1 acquisition in participants with high V1V2-binding antibodies [7,8].

A non-human primate (NHP) vaccine study which followed up on the results of the RV144 trial with an ALVAC prime and a Pentavalent B/E/E/E/E gp120 boost previously showed that improved coverage of V2 epitopes in the vaccine can improve protection (55%) against a heterologous Tier-2 subtype E/C SHIV challenge, compared to the RV144 ALVAC prime-B/E gp120 boost vaccine regimen which showed no protection [9]. Given that the improved coverage of V2 resulted in higher levels of protection together with the substantially higher diversity of subtype C viruses in southern Africa compared to CRF01_AE strains in Thailand, there is a stronger rationale to increase the diversity of immunogens included in subtype C vaccines [10].

In this study, we performed an NHP vaccine study to determine whether an ALVAC prime with gp120 boost immunogens, computationally designed to maximize subtype C V1V2 coverage and including signatures associated with sensitivity to broadly neutralizing antibodies can improve the protection of NHPs from a heterologous subtype C SHIV challenge compared to ALVAC prime-gp120 boost vaccines similar to the RV144 regimen and the P5 (Pox-Protein Public–Private Partnership) subtype C version tested in HVTN100 and HVTN702 [5,10,11]. Humoral immunogenicity of the four vaccine regimens was defined and five immune parameters were pre-selected for their correlation with protection. We then used a systems serology approach to determine whether immune markers could predict protection from infection regardless of vaccine regimen.

## 2. Methods

### 2.1. Ethics Statement

Rhesus macaques (M. mulatta) were housed at Bioqual, Inc. (Rockville, MD, USA), in accordance with the standards of the American Association for Accreditation of Laboratory Animal Care. The protocol was approved by Bioqual’s Institutional Animal care and Use Committee under OLAW Assurance Number A-3086-01. Bioqual is IAAALAC-accredited. This study was carried out in strict accordance with the recommendations in the Guide for the Care and Use of Laboratory Animals of the National Institutes of Health (NIH) and with the recommendations of the Weatherall report: ‘The use of non-human primates in research’. All procedures were performed under anesthesia using ketamine hydrochloride, and all efforts were made to minimize stress, improve housing conditions, and provide enrichment opportunities (for example, social housing when possible, objects to manipulate in cages, varied food supplements, foraging and task-oriented feeding methods, interaction with caregivers and research staff). Animals were euthanized by sodium pentobarbital injection in accordance with the recommendations of the panel on Euthanasia of the American Veterinary Medical Association.

Human PBMCs from HIV-1-negative individuals were collected with Institutional Review Board approval by the Duke Medicine Institutional Review Board for Clinical Investigations (protocols Pro00000873 and Pro00009459). All subjects consented following 45 CFR 46 and written informed consent was obtained by all participants. No minors were recruited into this study.

### 2.2. Vaccine Design and Production

The vaccine consisted of two primes with ALVAC-HIV (vCP2438) (previously described in [11]) and five boosts of ALVAC-HIV with different combinations of gp120 proteins.

Three natural subtype C Env proteins were selected for delivery, based on providing the best complementary coverage of the V2 region among known subtype C sequences [10]. These were combined with proteins used in the P5 (HVTN100 and HVTN702 trials; 1086c and TV1.21) [11] and RV144 Thai trial (A244_AE) [1]. Sequences for the three newly selected strains are available in GenBank as CAP260: JN681228, CAP174: JN967791, and Ko224: JN681240. Gp120 recombinant proteins used for immunization were produced by the Duke Human Vaccine Institute Protein Production Facility, Duke University Medical Center, as previously described [12].

### 2.3. Immunization and SHIV Challenge of Rhesus Macaques

Forty-five adult Indian origin male and female rhesus monkeys (*Macaca mulatta*) were genotyped and animals with the protective major histocompatibility complex class I alleles Mamu-A*01, B*08, and B*17 were divided equally among groups. Monkeys were housed at Bioqual, Rockville, MD. The animals were maintained in accordance with the National Institutes of Health and Harvard Medical School guidelines and all studies were approved by the appropriate Institutional Animal Care and Use Committee. Selected animals were randomly assigned to each vaccine arm (n = 9), such that each group had an equal number of male and female monkeys with equal mean ages and weights across groups. Monkeys were vaccinated by the intramuscular (IM) route with 1 × 10^8^ pfu ALVAC (vCP2438; Sanofi Pasteur, Lyon, France) vector alone twice and then boosted five times with 100 µg total purified Env gp120 protein in ALFQ adjuvant. ALVAC was delivered IM into one leg and the protein was delivered IM into the other leg. Immunized and control animals were then challenged six times weekly by the intrarectal route with an SHIV-CH505.375H.dCT challenge stock. The virus stock was grown from the infectious molecular clone in rhesus peripheral blood mononuclear cells (PBMCs) and the stock was titrated in rhesus macaques to select the appropriate dilution. Sequence diversity of the viral stock was determined by single-genome amplification [13]. Immunizations and challenges were carried out over three days and divided across groups to minimize confounding due to the day of immunization or challenge.

SHIV plasma viral RNA measurements were performed at the Immunology Virology Quality Assessment Center Laboratory Shared Resource, Duke Human Vaccine Institute, Durham, NC, as described [13].

### 2.4. HIV-1-Specific Binding Antibody Multiplex Assay (BAMA)

HIV-1-specific IgG antibodies to gp120 proteins and V1/V2 scaffolds were measured at the time of challenge (week 143) by an HIV-1 BAMA as previously described [2,14], with the exception that the detection antibody was biotinylated goat anti-monkey IgG (Rockland Immunochemicals, Inc., Pottstown, PA, USA). Positive controls included an HIVIG and CH58 monoclonal antibody IgG titration. Negative controls included in every assay were blank (uncoupled) and MulVgp70_His6 (empty gp70 scaffold) coupled beads, with a blank well on each assay plate, as well as HIV-1 negative sera. To control for antigen performance, we used the preset criteria that the positive control titer (HIVIG) included on each assay (and for assays with V1V2 antigens, CH58 monoclonal antibody [3]), which had to be within +/− 3 standard deviations of the mean for each antigen (tracked with a Levey–Jennings plot with preset acceptance of titer and calculated with a four-parameter logistic equation). Antibody measurements were acquired on a Bio-Plex 200 instrument (Bio-Rad, Hercules, CA, USA) using 21CFR Part 11 compliant software and the readout was in Median Fluorescence Intensity (MFI). All samples were tested at a screening dilution of 1:80 and post baseline sample testing was followed by a 5-fold serial dilution for a total of six dilution points (1:80 to 1:250,000). Area under the curve (AUC) of titrations was calculated for each sample and antigen using the trapezoidal rule, where the *x*-axis is log10 of dilution and *y*-axis is truncated MFI with negative values set to 0 for the calculation. If one or more dilutions were missing due to incomplete titration or failing QC, AUC was not calculated. Positivity of plasma binding response was determined at the screening dilution (1:80) with the following positivity criteria: (1) MFI > antigen-specific cutoff (which is 95th percentile MFI of all baseline plasma samples and 100 minimum); (2) MFI > 3-fold MFI of the matched baseline sample before and after subtracting the negative control bead MFI.

### 2.5. Magnitude-Breadth Score Calculation

The magnitude-breadth of responses was evaluated for binding to 14 heterologous V1V2 antigens designed to calculate the breadth of V1V2 responses [11]. Two of the V1V2-breadth antigens were not included as they matched vaccine proteins. A magnitude-breadth score is calculated for each animal as the area under the magnitude-breadth curve (AUC-MB), where the breadth represents the proportion of V1V2 antigens for which the animal had a positive vaccine-induced response at each magnitude interval, and AUC-MB is equivalent to the mean of the log10 scaled response magnitudes over the V1V2 antigen panel and available when all panel antigens’ data were available. The plots of partial MB curves included both individual subject-specific and group-specific curves.

### 2.6. ADCC Against gp120-Coated Target Cells

The GranToxiLux assay was used to detect ADCC activities of NHP plasma samples directed against CEM.NKRCCR5 CD4. T cells (NIH AIDS Reagent Program Division of AIDS, NIAID, NIH from Alexandra Trkola, University of Zurich, Zurich, Switzerland) were coated with recombinant gp120, as described [15]. ADCC activities of four-fold serial plasma dilutions starting at 1:100 were measured against cells coated with gp120 isolates representing vaccine immunogens A244.AE, 1086.C, CAP260.C, CAP174.C, and Ko244.C and the challenge virus CH505.C. Cryopreserved human PBMCs from an HIV-seronegative donor (Duke University, Durham, NC, USA) with the heterozygous 158F/V genotype for Fc-gamma receptor IIIa were used as the source of effector cells [16]. Data were reported as the maximum proportion of cells positive for proteolytically active granzyme B out of the total viable target cell population (maximum %GzB activity) after subtracting the background activity observed in wells containing effector and target cells in the absence of plasma. ADCC end point titers (ADCC titer) were determined by interpolating the dilutions of plasma that intercept the previously established positive cutoff for this assay (8% GzB activity) using GraphPad Prism, version 6.0f software (GraphPad Software, Inc., San Diego, CA, USA). ADCC assays were performed in duplicate for each animal at each time point.

### 2.7. Antibody-Dependent Cellular Phagocytosis Assay

Antibody-dependent phagocytosis was assessed by the measurement of the uptake of antibody-opsonized, antigen-coated fluorescent beads by the monocytic THP-1 cell line (ATCC; #TIB-201) [17]. Briefly, THP-1 cells were purchased from ATCC and cultured as recommended. Antibody-mediated phagocytosis assay was performed as described [18], with the following modifications. Briefly, 9 × 10^5^ beads coupled with CH505 gp120 Env were mixed with 10 µL of a 1:50 dilution of rhesus serum in a 96-well round-bottom plate and incubated at 37 °C for 2 h. During the incubation, blocking of CD4 on the cells was achieved by pretreating the cells at 10 × 10^6^ cells/mL with 20 µg/mL anti-human CD4 antibody (clone SK3) (Biolegend, San Diego, CA, USA) for 15 min at 4 °C. After incubation at 37 °C for 2 h, 2.5 × 10^4^ THP-1 cells were added to each well with a final volume of 100 µL each, then spinoculated at 1200× *g* for 1 h at 4 °C. Following spinoculation, immune complexes and cells were incubated at 37 °C for 1 h to initiate phagocytosis internalization. Cells were then fixed in 2% paraformaldehyde. Phagocytosis score was calculated by sample-specific percentage of positive cells x mean MFI, normalized to the corresponding result for the no-antibody control. A background level of phagocytosis was determined based on the mean + 3 s.d. of non-HIV-specific antibodies. Assays were performed in duplicate for each animal and the average phagocytosis score over the replicates was reported.

### 2.8. Monoclonal Antibody Competition ELISAs

Monoclonal antibody competition assays were performed as described previously [9,19]. In brief, plates were coated with HIV-1 envelope protein, washed, and blocked. Non-biotinylated monoclonal antibodies were serially incubated in triplicate wells for 90 min, followed by the addition of biotinylated monoclonal antibodies at sub-saturating concentrations and a 1 h incubation. The same antibody, in non-biotinylated form, was used to block itself as a positive control. An anti-influenza antibody CH65 was included as a negative control. Binding of biotinylated monoclonal antibodies was determined with horseradish peroxidase (HRP)-conjugated streptavidin. Binding of the biotinylated monoclonal antibody to HIV-1 envelope in the presence and absence of competing antibody was compared to calculate the percent inhibition of binding. Historical data on the negative controls of the assay were tracked and an assay was considered valid if the positive control antibodies blocked greater than 20% of the biotinylated antibody binding.

### 2.9. Neutralization

Neutralization activities of animal plasma and purified antibodies were determined by the TZM-bl-cell-based neutralization assay [20]. Tier phenotyping of the pseudoviruses was assayed by sensitivity to a pool of HIV-infected serum as described [21]. Neutralization assays were performed in technical triplicate for all animals at each serum time point or antibody concentration.

### 2.10. Machine Learning

#### 2.10.1. Missing Data

V1V2 MB-AUC values were missing for four animals. Values were imputed using the average value for the respective group to ensure all immune features could be used in the model construction.

#### 2.10.2. Dimensionality Reduction

Dimensionality reduction using principal component analysis (PCA) was performed on the measured immune features (n = 36). Immune features were scaled and centered and PCA was performed using R package ‘stats’ [22]. The R package ‘ggplot’ was used to make PC biplots [23].

### 2.11. Group Classification

A 5-fold cross-validated random forest algorithm was trained and tested on the immune feature dataset (n = 36) using R package ‘randomForest’ [24]. Training and testing were repeated 100 times with different splits of animals. As a control, permutation testing, in which vaccine group labels were shuffled randomly and the same 5-fold cross-validated random forest model was used for training and testing, was performed and repeated 100 times to generate a null result distribution. Model accuracy was measured using Mathew’s Correlation Coefficient (MCC) and performance differences between the actual and permuted models were evaluated using statistical significance (Kolmogorov–Smirnov test) and effect size (Cliff’s delta), using packages ‘stats’1 and ‘effsize’ [25]. To evaluate the outcome of the classification model, a confusion matrix was constructed by converting the predictions into percentages across the 100 repeats. Feature contributions were averaged and reported along with standard deviation across the 100 repeats.

### 2.12. Classification of Animals According to Susceptibility to Infection

Based on the number of challenges to infection, animals were divided into two groups: (i) susceptible to infection (≤4 challenges, n = 21) and (ii) resistant to infection (>4 challenges, n = 14). As described above, a 5-fold cross-validated random forest algorithm was trained and tested on the immune feature dataset.

### 2.13. Survival Analysis Using Predictive Models

As previously reported [9,26,27], a multivariate survival analysis approach was used to train a Cox proportional hazards model to predict the risk of infection at each challenge time point using Fc array features at 143 weeks post challenge. The steps involved in the analysis were as follows:

### 2.14. Model Training and Risk Prediction

Cox models were trained using the R package ‘survival’ [28]. Models were trained and tested using 10-fold cross-validation to predict risk of infection of each animal relative to the mean risk observed across all animals. Model performance was evaluated by Concordance Index [29]. The cross-validation was repeated 100 times with different splits of animals. As a control, permutation testing was performed using randomly shuffled challenge data and following the same exact procedure to filter features and train and test models. Performance differences between the actual and permuted data were characterized by measuring statistical significance and effect size (Cliff’s delta) on the Concordance Index distributions of the tested models’ actual and permuted data.

### 2.15. Representative Model Predicted Risk of Infection for Each Animal

A representative run of 10-fold cross-validated modeling was performed using the features that appeared in at least 50% of the models obtained over the repeated cross-validation. The representative run enabled the plotting of predicted risk for each animal when part of a test set.

### 2.16. ‘Risk Group’ Model to Compare Survival Probability of the Predicted and Susceptible Groups

For this purpose, a ‘risk group’ model was constructed. The survival probabilities for the resistant and susceptible groups were estimated by making a prediction of the mean values of features of animals in each group, i.e., ‘mean’ animal for the group. The predicted and actual Kaplan–Meier curves for the two groups were compared by log-rank statistics. We found no statistical evidence of difference between the actual and predictive curves.

### 2.17. Statistical Analyses

The number of NHPs for each group was calculated as the minimum required for at least 80% power using a log-rank test for comparison of survival curves. Investigators were blinded to the vaccine arm of the samples at the time of assay but were not blinded when analyzing the data. The Kaplan–Meier curves based on observed data were compared between groups using the log-rank test. Cox proportional hazards models were used for evaluating potential correlates of protection.

Group comparisons were made using exact Wilcoxon–Mann–Whitney tests. All statistical analysis was performed using SAS v9.4 (SAS Institute, Inc., Cary, NC, USA). GraphPad Prism version 6.01 was used for graphical representation.

### 2.18. Systems Serology Analysis and Code Availability

Systems serology analysis was performed using R version 4.2.3 in software Rstudio version 2023.12.1 + 402. The code used for dimensionality reduction and supervised machine learning using the random forest framework will be made available upon request. The code for survival analysis was adapted from Pittala et al., 2019 and Ackerman et al., 2018 [26,27].

## 3. Results

### 3.1. ALVAC Prime with Subtype C gp120 Trivalent Boost Protects from Repeated Intrarectal SHIV Challenge

To assess whether immunogens computationally selected to provide additional V1V2 coverage could improve vaccine efficacy against a heterologous SHIV, we immunized four groups (n = 9) of Indian origin rhesus macaques (RM; *Macaca mulata*) with an ALVAC vCP2438 (ALVAC C) prime and gp120 protein boosts in ALFQ adjuvant consisting of (i) A244.AE, 1086.C, CAP260.C, Ko244.C, and CAP174.C (group 1; Pentavalent), (ii) TV1.C and 1086.C (group 2; P5 Bivalent), (iii) A244.AE and 1086.C (group 3; RV144 Bivalent), or (iv) CAP260.C, CAP174.C, and Ko244.C (group 4; Trivalent) (Figure 1A). RM were primed with ALVAC twice at 0 and 4 weeks and then initially boosted three times at weeks 13, 21, and 47. Due to the COVID-19 pandemic, the study was halted and reinitiated a year later when two additional boosts were given at weeks 101 and 139. This was followed by a weekly low-dose intrarectal challenge with the heterologous Tier-2 transmitted/founder SHIV-CH505, starting at week 143 (Figure 1B). Of note, 66.7% (6/9) of RM from the Trivalent group remained uninfected after five challenges, whereas only 37.5% (3/8) from the Pentavalent group, 2/9 (22.2%) from the P5 Bivalent group, and 1/9 (11.1%) from the RV144 Bivalent group remained uninfected. All animals in the control group became infected by the fourth immunization. This resulted in overall statistically significant protection per exposure compared to the control group (*p* = 0.0225, log-rank test). When comparing groups individually to the control arm, protection remained significant in the Trivalent and Pentavalent groups (*p* = 0.0041 and *p* = 0.0260, respectively) (Figure 1F,C) but not in either Bivalent group (Figure 1D,E). Among animals that became infected, there was no impact on peak viral load compared to the control animals (Figure 1G).

### 3.2. Different gp120 Combination Boosts Elicit Similar Magnitudes of gp120 and Breadth of V1V2-Binding Antibodies

To understand how the different combinations of gp120 immunogens impacted responses in the different groups, we evaluated the immunogenicity of each vaccine at the time of challenge. We first measured binding antibodies to the six gp120 immunogens and a gp120 protein representing the envelope of the challenge virus, CH505TFD7gp120. Because of the limited number of animals, we did not adjust for multiple comparisons throughout but FDR-corrected *p*-values are reported in Appendix A. Responses to the gp120s were generally similar and of high magnitude among all four groups (Figure 2A). Without adjusting for multiple comparisons, the Trivalent group had significantly higher responses to several antigens, including higher responses to CAP174D11gp120 and CAP260D11gp120 than the RV144 Bivalent group (*p* = 0.0315 and 0.0056, respectively) and to A244D11gp120 and CAP260D11gp120 than the P5 Bivalent group (*p* = 0.0315 and *p* = 0.0106, respectively). The Pentavalent group also had significantly higher binding to A244D11gp120 than the P5 Bivalent group (*p* = 0.0055).

The breadth of V1V2 responses elicited by each vaccine at the time of challenge were also measured against a panel of 14 heterologous V1V2 gp70 proteins, representative of global HIV-1 diversity [30], which were used to generate area under magnitude and breadth curves (AUC-MBs) (Figure 2B). Comparing the AUC-MBs revealed similar breadth of responses across the four vaccine groups, except for the RV144 group, which had significantly lower V1V2 AUC-MBs than the P5 Bivalent group (*p* = 0.0055).

### 3.3. Vaccines Elicit Strong Functional Antibody Responses

To understand the functional capacity of the antibodies elicited by each vaccination at the time of challenge, we tested the ability of antibodies in sera at the time of challenge to mediate, or neutralize, Fc effector functions against, the immunogen or challenge envelopes. Among the groups, antibody-dependent cellular cytotoxicity (ADCC) responses tended to be highest in the Pentavalent group: of the six envelopes tested, significantly different responses were detected against 1086C.gp140, A244D11gp120, CAP174D11gp120, and Ko244D11gp120. Against the 1086C.gp120, the Pentavalent group (median pAUC = 4.8) had significantly higher levels of ADCC than the Trivalent (median pAUC = 0; *p* = 0.0137) and RV144 Bivalent (median pAUC = 0; *p* = 0.0123) groups. The Pentavalent group (median pAUC = 3.51) had significantly higher levels of ADCC against A244D11gp120 than the Trivalent group (median pAUC = 0; *p* = 0.0097). Against the CAP174D11gp120, the Pentavalent group and P5 Bivalent groups (median pAUCs = 4.94 and 4.82, respectively) had significantly higher levels of ADCC than the RV144 Bivalent group (median pAUC = 3.60; *p* = 0.0027 and 0.0196, respectively). Lastly, the Pentavalent group (median pAUC = 4.54) had significantly higher levels of ADCC against Ko244D11gp120 than the RV144 Bivalent group (median pAUC =4.12, *p* = 0.0283) (Figure 3A). There were no differences in magnitudes of ADCC against the envelope representing the challenge virus, CH505TFD7gp120 (median pAUCs 3.94–4.84) (Figure 3A). While antibody-dependent cellular phagocytosis (ADCP) responses against CH505TFD7gp120 at the time of challenge were similar (median ADCP scores: 17–18.95), the Trivalent group was significantly higher than the P5 Bivalent group (*p* = 0.0379) (Figure 3B).

Neutralization was only observed against easy-to-neutralize Tier-1 envelopes: the CRF01_AE envelope TH023.6 and the subtype C envelope TV1C8.2 (Figure 3C). Neutralization of TH023.6 was significantly lower in the P5 Bivalent group (median ID_50_ titer: 432) than the other three groups (median ID_50_ titers: 3581–4462, *p*-values: 0.0315–0.0010) and all four groups had similar neutralizing antibody responses to TV1C8.2 (median ID_50_ titers: 1450–2383). Very low (ID_50_ < 35) to no neutralization of the Tier-2 challenge virus, SHIV.CH505.375H.dCT, or the other five Tier-2 envelopes tested was detected (Appendix A).

We next assessed the capacity of antibodies in sera to block binding of soluble CD4 or several monoclonal antibodies to the gp120 envelopes A244D11gp120 (CD4, A32, CH58, PG9, CH01, and 2G12), 6240 (CD4, PG9), and 902114B2gp140 (CH01). The only parameters with significant differences between groups were blocking of CD4 binding to A244D11gp120, where blocking by antibodies in the P5 Bivalent group (median = 81.6%) was significantly lower than the other three groups (medians = 97.6–100%; *p* = values 0.0002–0.0077), blocking of the anti-V2 non-neutralizing monoclonal antibody CH58 binding to A244D11gp120, where blocking by antibodies in the Trivalent group (median = 24.1%) was significantly lower than the other three groups (medians = 90.2–95.9%; *p* = values 0.00008–0.0005) (Figure 3D), and blocking of 2G12 to A244D11gp120, where the Pentavalent group (median = 36.2%) was significantly higher than the P5 Bivalent group (median = 28.1%, *p* = 0.0037) (Appendix A). Interestingly, blocking of the V2-targeting broadly neutralizing antibody CH01 also showed a higher trend in the Trivalent (median = 63%) and Pentavalent (median = 56.9%) groups compared to the Bivalent RV144 (median = 44.5%) and P5 Bivalent (median = 39.2%) groups (Figure 3D).

### 3.4. ADCC Responses Against CH505 at the Time of Challenge Correlated with Protection from Infection

To identify immune correlates of protection, five immune response readouts at the time of challenge were pre-selected to conduct a Cox proportional hazards analysis, which considers infection status and time to infection together with immune marker levels. These consisted of ADCC, IgG-binding antibodies, and ADCP against the gp120 of the challenge virus, CH505TFD7gp120; blocking of CD4 binding to A244D11gp120; and V1V2-binding antibody MB-AUCs. A combined analysis of all groups together identified only ADCC responses against CH505TFD7gp120 as a correlate of protection at the time of challenge (hazard ratio: 0.793; 95%CI: 0.648–0.970, *p* = 0.0241) (Table 1). To confirm this result, we used a pairwise Cox model with the same immune responses for control vs. each experimental group. In this analysis, ADCC, IgG-binding antibodies, and ADCP responses against CH505TFD7gp120 and V1V2 MB-AUCs were all significantly associated with protection in both the Pentavalent and Trivalent groups (Appendix A).

### 3.5. Immune Responses Induced by the Different Vaccine Arms

As an unbiased way to evaluate differences in immune responses in immunized animals, we performed unsupervised dimensionality reduction using principal component (PC) analysis. Because V1V2 MB-AUCs for four animals were missing, and these analyses cannot be completed without the full dataset, values for those animals were imputed by inserting the average MB-AUC value of the relevant groups. PC biplots demonstrated that animals in the Trivalent group, the group that showed the greatest degree of protection, were distinct in PC2 values, as compared to the animals in the Pentavalent, P5 Bivalent, and RV144 Bivalent groups (Figure 4A). Blocking CH58 from binding to the CRF01_AE gp120 A244D11gp120 and ADCC against Ko224D11gp120- and A244D11gp120-coated cells were the top three features that made the strongest contribution to PC2 (Appendix A), in line with the fact that the Trivalent group was the only group not to include the A244.AE/1086.C V1V2 linear epitope. While these features differentiate the immunological profiles of the vaccine groups based on our test reagents, it worth emphasizing that group differentiation driven by the evaluation of a CRF01_AE virus as a test reagent (A244D11gp120) may have limited implications for vaccine breadth within southern Africa. Group 4, which ranked highest in terms of protecting against a C clade heterologous challenge, did not include A244 or any other AE vaccine in the vaccine cocktail.

To identify signatures of the responses that were specific to each of the different vaccine arms, we then constructed a repeated 5-fold cross-validated supervised classifier using the random forest algorithm. Model accuracy for test data across folds and replicates was calculated using Mathew’s Correlation Coefficient (MCC) and demonstrated high accuracy in classifying animals according to vaccine arm. Comparison of MCC values yielded for models trained on actual versus permuted class labels indicated that these predictions were robust; the high quality of classification results was unlikely to be observed at random and demonstrated a large effect size, as measured by Cliff’s delta (Figure 4B). To explore classification accuracy observed for each individual class, a confusion matrix was plotted, demonstrating that the model accurately classified animals according to vaccine arms with an overall accuracy of ~67% (Figure 4C). The Trivalent group (group 4), which showed distinct immunological responses from the other groups, was classified correctly >79% of time while the Pentavalent and two Bivalent groups (groups 1–3) were classified with similar accuracy (57–68%). The 25 immune features that contribute the most to group classification using the random forest model included blocking of CH58 and CD4 binding to A244D11gp120 and breadth of V1V2-binding antibody responses (Figure 4D). Overall, supervised models identified characteristic differences in immunogenicity data between immunization groups.

### 3.6. Immune Correlates of Decreased Infection Risk

We then divided animals into two groups based on susceptibility to infection, to study protection from infection irrespective of vaccine immunogen design. Animals that were infected in ≤4 challenges were defined as susceptible (n = 21) and those infected at challenge 5 or those that remained uninfected were defined as resistant (n = 14). In terms of PC, susceptibility groups showed some evidence of distinct profiles (Appendix A), and when a repeated 5-fold cross-validated random forest classifier was trained and tested, accurate and robust classification was observed (Appendix A). The confusion matrix demonstrated accurate classification of animals belonging to resistant and susceptible groups >66% of the time (Appendix A). The 25 immune features contributing the most to the random forest model included blocking of several proteins (CH58, CH01, and CD4) binding to various gp120s, IgG binding to CAP174D11gp120, and ADCC against cells coated with the challenge protein (CH505TFD11gp120) (Appendix A).

To define immune responses associated with reduced risk of infection with a greater degree of resolution than that afforded by classification, Cox proportional hazards (Cox) models were trained and tested following aggressive filtering and in the context of repeated cross-validation. Model robustness was demonstrated by both reproducible results across repeated cross-validation replicates and as compared to predictions made following risk class label permutation testing (Figure 5A). The agreement of predictions to observed risk of infection, as defined by Concordance Index, was significantly greater for models trained on actual than on permuted risk data, as demonstrated by the Kolmogorov–Smirnov test and high Cliff’s delta. Moreover, as demonstrated in a representative run, prediction of susceptibility to infection of individual animals was also accurate, with animals infected at earlier challenges exhibiting higher predicted risk (Figure 5B). When the animals were grouped based on the relative risk groups defined previously, animals in the susceptible group had higher predicted risk as compared to animals in the resistant group (Figure 5C). Indeed, when modeled based on group-wise immunogenicity profiles, the risk of infection for resistant and susceptible groups predicted by the model closely mirrored the actual infection risk as represented in Kaplan–Meier curves (Figure 5D). The final model employed a set of seven immune features that predicted susceptibility to infection (Figure 5E). Of those, blocking of PG9 binding to the CRF01_AE gp120 A244D11gp120 was significantly associated with susceptibility to infection and blocking of CH01 to the subtype B gp140 902114B2gp140 and ADCC against 1086Cgp140-coated cells were significantly associated with resistance to infection. Overall, these results indicate that robust relationships exist between immunogenicity data and challenge outcomes, and sera antibodies that were able to block different V2 monoclonal antibodies from binding contributed to the prediction of either susceptibility or resistance to infection.

## 4. Discussion

In this study, we tested combinations of HIV immunogens computationally selected to improve the coverage of subtype C V1V2 diversity in a rigorous NHP challenge model to determine whether improved coverage could increase vaccine efficacy against a subtype C heterologous challenge. All groups elicited generally similar levels of humoral responses pre-challenge and low-to-no detectable Tier-2 neutralizing antibodies. However, the Trivalent and Pentavalent groups were both able to significantly protect animals from infection with the difficult-to-neutralize (Tier-2) challenge SHIV-CH505TF.375H.dCT, with survival rates of 66.7% and 37.5%, respectively. A combined analysis of all groups together identified only ADCC responses against CH505TFD7gp120 as a correlate of protection at the time of challenge (hazard ratio: 0.793; 95%CI: 0.648–0.970, *p* = 0.0241). Applying a machine learning systems serology approach showed that levels of blocking of V2 broadly neutralizing antibodies to different envelopes were able to predict susceptibility or resistance to infection across all groups.

The failure of the phase-IIb/III HVTN702 vaccine efficacy trial (the southern African follow-up trial to the RV144 trial) to provide any protection in southern Africa was disappointing for the field [5]. There are several hypotheses for this failure, including the significantly increased diversity of subtype C viruses in southern Africa relative to CRF01_AE viruses in Thailand [10]. To address this issue, we designed vaccine regimens to maximize the coverage of diversity observed in the highly variable V1V2 region of subtype C viruses given the practical constraints of using no more than three natural variants in a vaccine cocktail [10]. Here, we compared vaccine responses in NHPs using the three optimized variants we selected for C clade coverage (group 4): a Bivalent vaccine that included 2 C clade variants that were not particularly selected with respect to natural C clade diversity coverage (group 2); a Bivalent vaccine that included one C clade but also the original CRF01 AE variant A244.AE to address the possibility that this Env might intrinsically favor V1V2 directed responses and this may have contributed to the protection observed in the RV144 trial (group 3); and one Pentavalent vaccine that combined groups 3 and 4. The groups boosted with the computationally selected envelopes were significantly more protected from a heterologous challenge than the groups that did not include them (66.7% and 37.5% versus 22.2% and 11.1%), suggesting that this approach succeeded in this study.

Interestingly, when considering the V2 sequence of the vaccine regimens to the challenge envelope, the Trivalent and Pentavalent include the immunogen CAP260.C gp120, which contains an Arginine at position 169 (R169) similar to the challenge envelope CH505, while the other regimens included immunogens which all contained a Lysine (K169). Position 169 is of particular interest because a match in this position was the only V2 signature site for which a match to the vaccine was positively associated with vaccine efficacy in the RV144 trial [31]. It is also a highly variable position in the HIV envelope: within clade C, K169 is present in 67% of envelope sequences, followed by R169 (8%), and a mixture of rarer amino acids make up the remaining 25% (N169 is present in ~3% of envelopes). The Trivalent and Pentavalent vaccine regimens carry mixtures of K169, R169, and N169; therefore, it is possible that diversity in this position in the vaccine cocktail favored selection of antibodies that tolerate variation in position 169 and so do not require a specific match.

Here, a Cox proportional hazards primary correlates analysis associated ADCC responses against the challenge envelope at the time of challenge with protection from infection. ADCC was a correlate with decreased risk of infection in RV144 [2]. In RV144, genetic signatures in the V2 were associated with increased vaccine efficacy [31] and antibodies targeting a heterogenous site in the V2 were isolated from the sera of vaccinees [32]. In addition, a follow-up NHP study that aimed to improve the results of RV144 [9] and a DNA-protein coadministration NHP study [33] also found that ADCC was correlated with increased resistance to infection. These studies suggest that while broad neutralizing antibodies are the gold standard for an HIV-1 vaccine, effector functions of antibodies targeting this region may still play a role in protection.

Importantly, while ADCC was the primary correlate of protection in the traditional Cox analysis, the systems serology analysis revealed other factors that contributed to protection. In particular, blocking of V2 broadly neutralizing antibodies that recognize conformational epitopes binding to gp120s was associated with both increased susceptibility and resistance to infection; blocking of PG9 binding to A244D11gp120 was associated with susceptibility to infection while blocking of CH01 to 902114B2gp140 was associated with resistance to infection and blocking of CH01 to A244D11gp120 was highest in the Pentavalent and Trivalent groups, suggesting subtle differences in the epitopes of monoclonal antibodies elicited by the different vaccine arms. PG9 and CH01 both bind to conformational epitopes on the V2 loop but CH01 binds a distinct epitope/approaches the epitope from a different angle to PG9 [34]. Antibodies targeting a similar conformational epitope to CH01 but not PG9 may have played a role in protection in this study.

Analysis of the blocking of various proteins binding to A244D11gp120 allows some deconvolution of the responses developed in the Trivalent group. Sera in all groups were able to block CD4 binding to A244D11gp120, suggesting the development of antibodies that could recognize A244D11gp120 in all four groups regardless of the presence of the gp120 in the vaccine arm. However, sera from animals in the Trivalent group blocked the binding of CH58, which recognizes a linear V2 epitope [32], significantly less than the other three groups. This was the feature that separated the immune response in the Trivalent group from the other three groups the most. Instead, sera in this group were able to block CH01 binding to the highest extent, suggesting that immunogens in the Trivalent group were able to focus on B cell maturation and antibody development to preferentially target conformational V2 epitopes—a desired outcome of an HIV vaccine.

This study elicited antibodies that were able to neutralize Tier-1 envelopes but not the heterologous Tier-2 envelope used in the challenge or the five other Tier-2 envelopes representing the vaccine immunogens, suggesting that only weak neutralizing antibodies were elicited. This was not surprising given the use of gp120 proteins as immunogens. Despite this, the systems serology suggests that responses may be targeting a relevant epitope and alternative forms of the immunogen (i.e., trimeric forms of the envelope instead of monomers) may be able to elicit neutralizing antibodies targeting conformational V2 epitopes. However, further analysis of the B cell repertoire and monoclonal antibody isolation are needed to confirm the epitopes of antibodies elicited by these vaccines.

One interesting and unintentional feature of this study was the delayed period between the fifth and sixth vaccination due to the interruption caused by the COVID-19 pandemic. To our knowledge, this is the third NHP challenge study that included a significantly longer interval during the boost administration. In the ALVAC-AE Pentavalent gp120 boost study by Bradley et al. [9], there was a 41-week interval between the fourth and fifth doses, after which animals were challenged with SHIV-1157(QNE)Y173H; in a DNA + protein coadministration study by Felber et al., there was a 20-week interval between the fifth and sixth doses, after which animals were challenged with SHIV-CH505.375H [33]. Both studies also showed high levels of protection (55% and 67%, respectively). Collectively, these studies raise questions about the optimal period between boosts and suggest that a delayed boost may improve the quality of humoral immune responses. This needs to be explored further in future studies.

In conclusion, we have demonstrated that a vaccine with computationally selected immunogens to maximize coverage of the subtype C V1V2 region is able to protect NHPs from a heterologous SHIV challenge. Delayed boosting with computationally defined subtype C immunogens should be further investigated in clinical trials as a potential mechanism to induce protective antibody responses.

## Figures and Tables

**Figure 1 vaccines-13-00231-f001:**
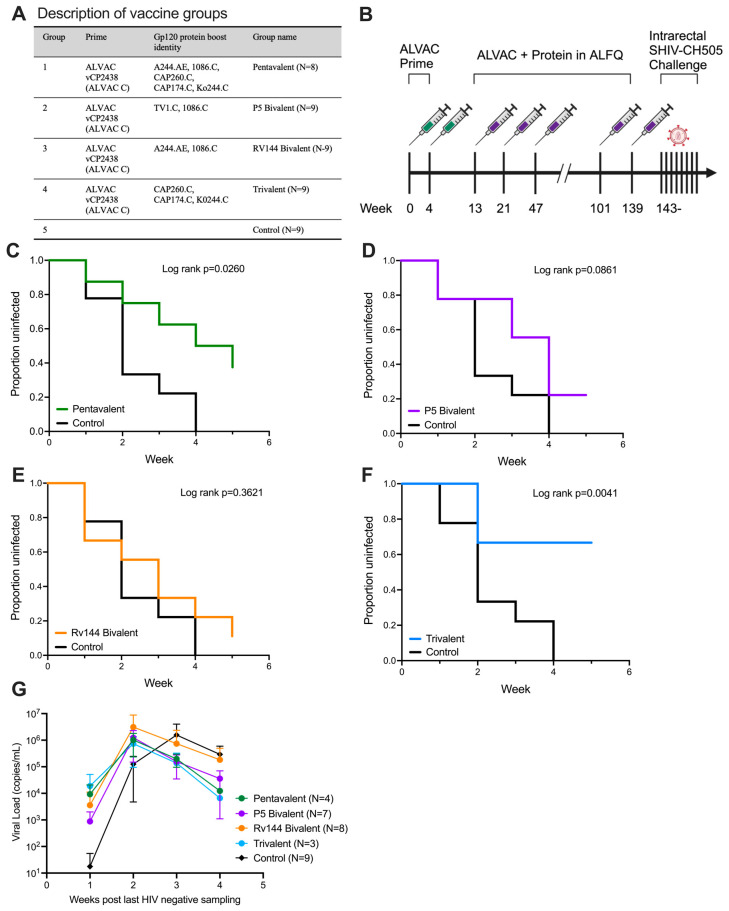
**ALVAC vCP2438 prime with a Trivalent gp120 boost protects NHPs from intrarectal challenge with 66% efficacy.** (**A**) Forty-four animals were divided into five groups and given one of four vaccine formulations or phosphate-buffered saline as a control. (**B**) All animals were administered two doses of ALVAC vCP2438 (ALVAC-CO four weeks apart and then animals received five doses of ALVAC-C with a Pentavalent, P5 Bivalent, RV144 Bivalent, or Trivalent protein boost in ALFQ adjuvant five times). All animals were then subjected to five weekly low-dose intrarectal challenges with SHIV-CH505.375H.dCT. (**C**–**F**) Kaplan–Meier plot showing the percentage of uninfected animals after five weekly challenges. Survival was significantly different among all vaccine groups compared to the control (*p* = 0.0225; one-tailed KM log-rank test). (**G**) Viral loads were tested weekly in all groups. Lines are the group means for animals that became infected and bars are the 95% confident intervals.

**Figure 2 vaccines-13-00231-f002:**
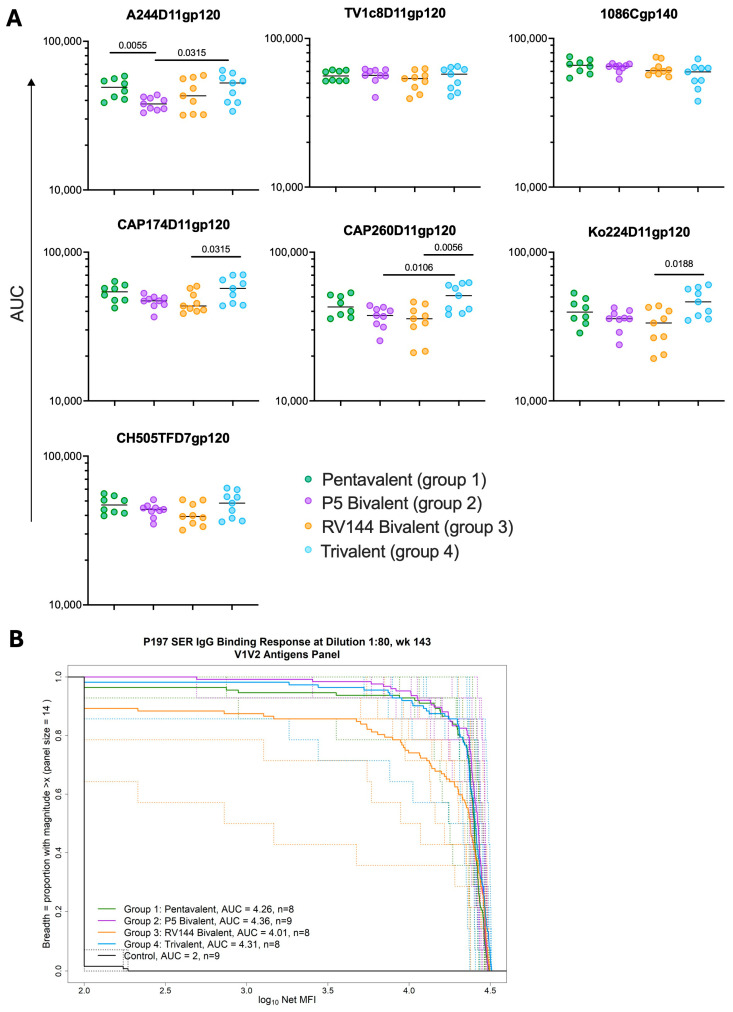
**Different gp120 combination boosts elicit similar magnitudes of gp120 and breadth of V1V2-binding antibodies.** (**A**) Binding antibodies against the vaccine and challenge gp120s were measured. (**B**) Fourteen V1V2 antigens were used to calculate the individual and the mean area under the magnitude breadth curve (MB-AUC) for each vaccine arm. Dotted lines are individual curves by NHP and the unbroken thicker lines represent the mean AUC-MB of the respective group.

**Figure 3 vaccines-13-00231-f003:**
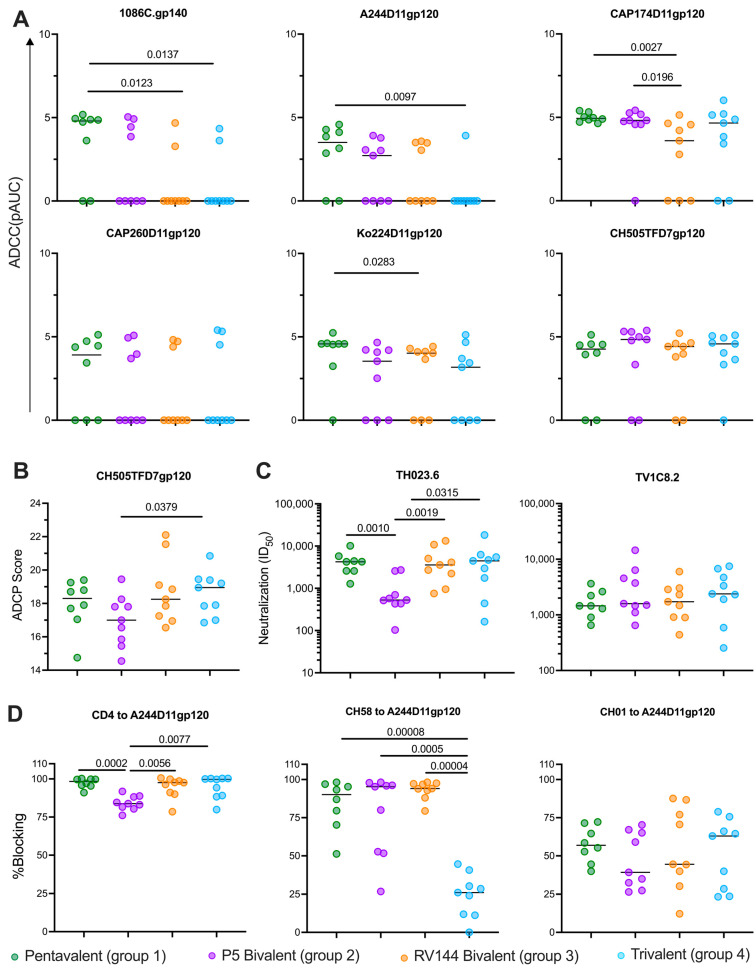
**Vaccine arms elicit similarly high functional antibody responses.** Vaccine-elicited (**A**) ADCC responses against cells coated with each of the five vaccines and challenge gp120, (**B**) ADCP of the challenge protein, (**C**) neutralization of two Tier-1 pseudoviruses, and (**D**) blocking of three proteins (CD4, CH58 mAb, and CH01 mAb) binding to the CRF01_AE protein A244D11gp120.

**Figure 4 vaccines-13-00231-f004:**
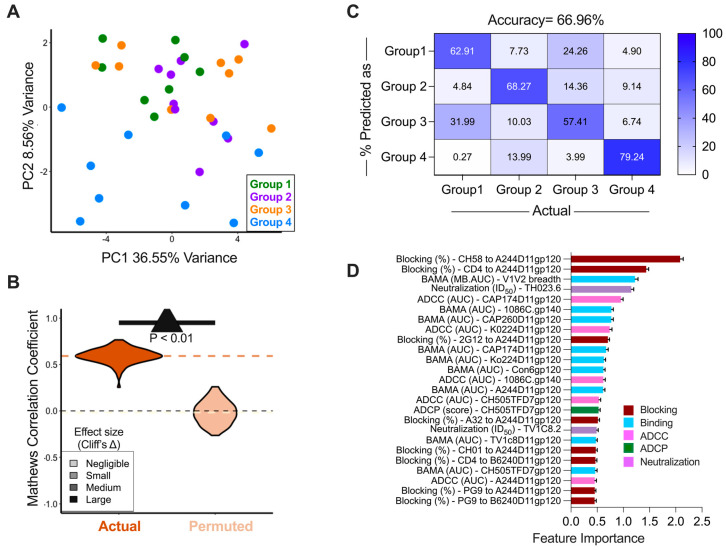
**Immune feature profiles robustly distinguish different arms of vaccination.** (**A**) Principal component (PC) biplot. Animals are represented as dots, with color indicating the vaccine group. (**B**) Comparison of Matthew’s Correlation Coefficient (MCC) as a measure of classification accuracy of vaccine group using random forest approach in 100 repeats of 5-fold cross-validation with actual and permuted group labels. Dotted orange line indicates the mean MCC of model with actual labels while dotted yellow line indicates mean MCC of model with permuted labels. Dotted black line indicates MCC = 0. Models trained on actual data outperform models trained on permuted data significantly (Kolmogorov–Smirnov test, inset) and substantially (Cliff’s delta, large effect size). (**C**) Confusion matrix depicting the percentage of animals classified into different groups across 100 repeats and 5-fold cross-validation using random forest approach. (**D**) Average feature importance of the top 25 features that contributed the most in building the classifier across 100 repeats. Error bars represent standard deviation across repeats.

**Figure 5 vaccines-13-00231-f005:**
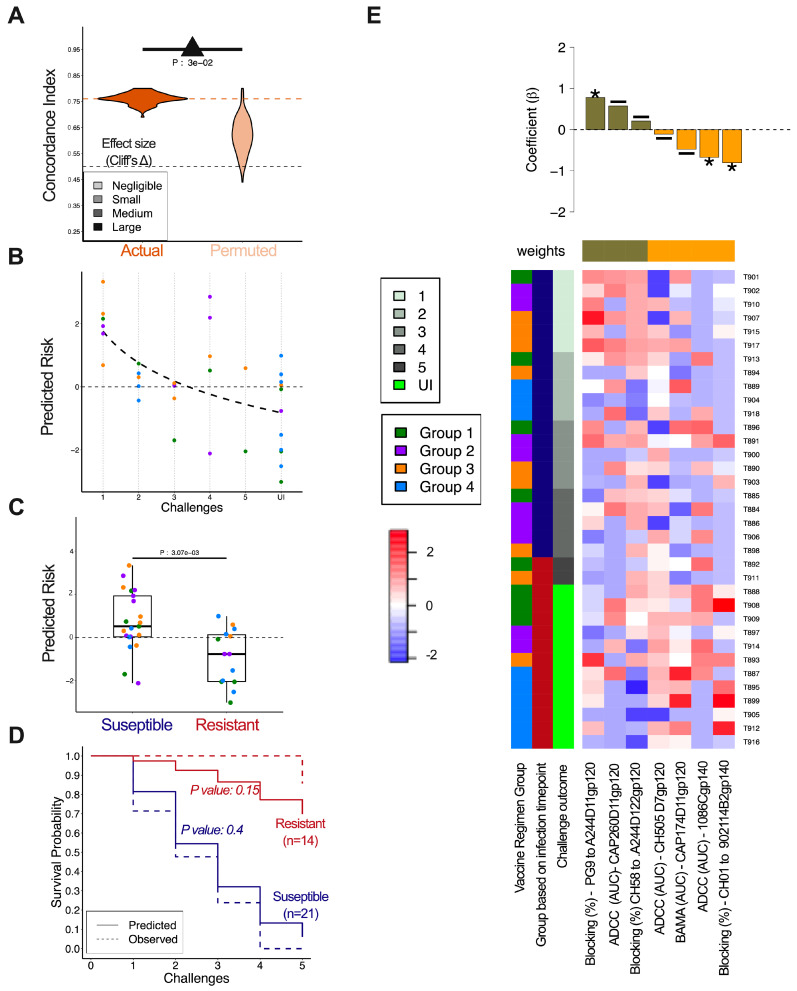
**Immune feature profiles robustly predict protection.** (**A**) Accuracy (Concordance Index) of risk predictions using actual versus permuted challenge outcome data across modeling replicates. Dotted orange and yellow lines indicate distribution means for actual and permuted dataset. Dotted black line indicates performance expected at random (0.0). Statistical significance was defined by Kolmogorov–Smirnov test (inset) and effect size by Cliff’s delta (color). (**B**) Scatterplot depicting the predicted relative risk of infection for each animal versus the challenge at which infection was observed for the representative model. The thick dotted line depicts the trendline. (**C**) Box plot showing the predicted relative risk of infection for animals in susceptible and resistant groups. Statistical significance was determined by Wilcoxon–Mann–Whitney test. (**D**) Kaplan–Meier graphs of observed (dotted line) and predicted risk group model (Cox PH, solid line) challenge outcomes learned from immune profiles for susceptible (blue, ≤4 challenges) and resistant (red, ≥5 challenges) groups. Significance between predicted and actual curves was defined by two-sided log-rank test. (**E**) Heatmap and feature coefficient plot of the filtered set of features (columns) and their contribution to the ‘final’ model (coefficients in bars; top panel). (Cox PH *p*-values: * *p* < 0.05; - *p* ≥ 0.05). Individual animals (rows) are ordered by time to infection and colored by vaccine group, relative susceptibility group, and time to infection (challenges). Features are centered and scaled with high responses indicated in red and low responses in blue.

**Table 1 vaccines-13-00231-t001:** Cox proportional hazard ratios for primary outcomes at time of challenge (week 143), n = 44.

Immune Marker	Antigen	Hazard Ratio (95% CI)	*p*-Value
IgG-binding antibodies	CH505TFD7gp120	0.5530 (0.060–5.125)	0.6020
V1V2-binding antibodies	V1V2 breadth (MB-AUC)	2.91 (0.27–31.69)	0.3808
ADCC	CH505TFD7gp120	0.793 (0.648–0.970)	0.0241
ADCP	CH505TFD7gp120	1.1607 (1.161–1.504)	0.2595
CD4 Blocking	A244D11gp120	0.1547 (0–59569)	0.7761

## Data Availability

The raw data supporting the conclusions of this article will be made available by the authors on request.

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
