# Peer review of "Computationally Selected Multivalent HIV-1 Subtype C Vaccine Protects Against Heterologous SHIV Challenge"

_vaccines, 2025, doi:10.3390/vaccines13030231_

Round 1

Reviewer 1 Report

Comments and Suggestions for Authors

The manuscript by Mielke and colleagues describes a set of immunization experiments and subsequent analysis of correlates of protection in NHP challenged with an HIV-1 subtype C vaccine(s).  The immunization scheme was complex and even further complicated by a hiatus in the immunizations during the COVID-19 pandemic.  Two of the arms of the immunization protocol provided significant protection from a SHIV challenge. While the immunization protocols were too complex for practical use (ie development in humans), the work provides potentially useful information with regards to the correlates of protection, particularly regarding the role of V2 and of ADCC.

The manuscript is well-written and the results amply documented. A few suggestions to improve readability:

1.        When was the “susceptibility to infection” defined, before the start of the experiments or after the analysis was completed as indicated in the methods section.

2.        Figure S1 with the survival curves of each of the immunogens vs. control macaques would be better suited for the manuscript itself as it is key to the interpretation of the rest of the manuscript and the single graph is harder to read in the copy (as with many of the other graphs, eg 2A).

3.        Figure S4 is much easier to grasp and illustrates the model better than Fig 4 in the manuscript. The authors should consider a substitution. 

Author Response

Reviewer 1:

The manuscript by Mielke and colleagues describes a set of immunization experiments and subsequent analysis of correlates of protection in NHP challenged with an HIV-1 subtype C vaccine(s).  The immunization scheme was complex and even further complicated by a hiatus in the immunizations during the COVID-19 pandemic.  Two of the arms of the immunization protocol provided significant protection from a SHIV challenge. While the immunization protocols were too complex for practical use (ie development in humans), the work provides potentially useful information with regards to the correlates of protection, particularly regarding the role of V2 and of ADCC.

The manuscript is well-written and the results amply documented. A few suggestions to improve readability:

  1. When was the “susceptibility to infection” defined, before the start of the experiments or after the analysis was completed as indicated in the methods section.

Susceptibility to infection was defined using infection outcomes, which are only known after animals are challenged, as described in the methods (lines 541-544).

  1. Figure S1 with the survival curves of each of the immunogens vs. control macaques would be better suited for the manuscript itself as it is key to the interpretation of the rest of the manuscript and the single graph is harder to read in the copy (as with many of the other graphs, eg 2A).

We thank the reviewer for this suggestion and agree that figure S1 is more suited to interpret as a main figure. Consequently, we have replaced figure 1C with the individual survival curves. In addition, we increased the size of graphs of figure 2A to make them easier to visualize and interpret.

  1. Figure S4 is much easier to grasp and illustrates the model better than Fig 4 in the manuscript. The authors should consider a substitution.

There are two machine learning tasks described in this manuscript:

  • The first task describes modeling the extent to which and what differences in immune features distinguish between vaccine regimens and is visualized in Figure 4 (lines 218-219 and 233-234).
  • In contrast, Figures S4 (now S3) and 5 describe the second task, which was modeling the extent to which and what immune features distinguish animals with differing resistance to infection, which is presented in categorical (relatively resistant/sensitive - Figure S4) and continuous (number of challenges to infection – Figure 5) forms (lines 255-258, 265-267).

Consequently, Figure 4 and S4 cannot be substituted. While Figure S4 could replace Figure 5, the good performance achieved in modelling the more challenging task (continuous rather than categorical outcomes) motivated giving this analysis greater priority and inclusion as a main figure.

Reviewer 2 Report

Comments and Suggestions for Authors

Mielke et al perform a vaccine trial in NHPs using combined subtype C envelope proteins with high diversity as boosters to confer protection against SHIV challenge. Two of their vaccination groups exhibited significant differences in protection against challenge compared to several other groups included in the study. Overall, this is an interesting study with potentially impactful findings about future HIV-1 vaccine designs. I have some minor clarifying questions details below

Major

-              Unless I missed it, it is unclear how the immunogens were computationally selected and what the criteria were for exclusion/inclusion. The methods and results section don’t seem to elaborate on the selection process and the nature of the immunogens. I’m assuming this was how the boosting group compositions were assembled?

-              I understand the rationale for not conducting statistical tests using multiple comparisons because of low n-values when assessing antibody responses, but what do the comparisons look like if data are stratified by protected vs unprotected either across or within groups? For example, Fig. 2A has all animal values included.

-              In line with the comment above, do the highest ADCC responses in Figure 3 correlate with the animals that exhibited protection against challenge? Based on the gross analysis, it doesn’t seem as though the ADCC responses bolster protection since the bivalent group #2 respond the best but have poor protection

-              What is the amino acid diversity in the V1V2 loop region amongst the primary and booster Envs compared to the challenge Env? Does this explain the variability in protection phenotype across groups?

Minor

-              Table in Figure 1A is blurry/hard to read

Author Response

Reviewer 2:

Mielke et al perform a vaccine trial in NHPs using combined subtype C envelope proteins with high diversity as boosters to confer protection against SHIV challenge. Two of their vaccination groups exhibited significant differences in protection against challenge compared to several other groups included in the study. Overall, this is an interesting study with potentially impactful findings about future HIV-1 vaccine designs. I have some minor clarifying questions details below

Major

  1. Unless I missed it, it is unclear how the immunogens were computationally selected and what the criteria were for exclusion/inclusion. The methods and results section don’t seem to elaborate on the selection process and the nature of the immunogens. I’m assuming this was how the boosting group compositions were assembled?

The reviewer is correct that boosting compositions were computationally assembled using inclusion/exclusion criteria. This selection process is described in the recently accepted manuscript, Shen et al. which we referenced in lines 108-113 of the introduction. For the reviewer’s interest, we have extracted that portion of the manuscript and the relevant figure below:

“We computationally selected 3 Env sequences that can complement the three P5 subtype C vaccine strains for coverage of subtype C V1V2 sequence diversity. Previous studies found that longer V1V2 loop length is associated with resistance to autologous neutralization. Virus sensitivity for broadly neutralizing antibodies that target V1V2 quaternary epitopes is associated with glycosylation at specific residues, the length of the V1V2 region, and the net positive charge in the β sheet C region of V2. In addition, certain subtypes of HIV-1 select for viruses with shorter V1V2 loop during transmission. Therefore, in addition to coverage of V1V2 sequence diversity, we also selected for: 1) shorter V1V2 length; 2) positive net charge in V2 region; and 3) maintenance of key glycosylation sites.

First, we examined 119 acute subtype C sequences isolated from SA in LANL database for V1V2 hypervariable region length and net charge. A set of 17 sequences was selected that has V1V2 region net charge > 0, and total length < 32. Next, we selected the three sequences that provide the best complementary coverage of the V2 region to the existing subtype C vaccine strains. The three sequences selected were CAP 260, CAP 174, and Ko224.  We then confirmed that the key glycosylation sites N156 and N160 that are associated with sensitivity to quaternary V1V2 broadly neutralizing antibody sensitivity are present in the selected strains (Fig. 1B. Please refer to the PDF file for the Figure), and that they are not extreme outliers of the subtype C (Fig 1A). The subtype C phylogenetic tree (maximum likelihood built with FastTree) in Fig 1A shows where the vaccines are distributed relative to the C subtype globally.  About half of the SA sequences form a distinctive clade, and none of the 3 original P5 vaccine strains are a member of that clade. In contrast, two out of the three newly selected boost strains (CAP260 and CAP174) belong to that distinctive clade.  Subtype C Env diversity is well represented by the three newly selected boost strains plus the three original P5 subtype C strains even though the selection of these 3 novel strains was based on V1V2 region (Fig 1A). In addition, V1V2 sequence analysis showed excellent coverage of all the most common amino acids that were found in SA isolates in each position within the V1V2 epitope (Fig 1C, 1D), indicating that restricting the selection to shorter V1V2 length and positive charge did not compromise the coverage of V1V2 diversity.” Please refer to PDF file for the figure.

  1. I understand the rationale for not conducting statistical tests using multiple comparisons because of low n-values when assessing antibody responses, but what do the comparisons look like if data are stratified by protected vs unprotected either across or within groups? For example, Fig. 2A has all animal values included.

Figures 2 and 3 refer to the immunogenicity of the different vaccine regimens and do not relate to the protection status of the NHPs. Instead, this was analyzed using Cox Models, the results of which are summarized in Table 1. The analysis using a Cox Model uses time to infection to compare the hazard/risk of infection and were used to determine potential correlates of protection. For clarity, we have added the underlined sentence to the results (lines 205-207):

“To identify immune correlates of protection, five immune response readouts at the time of challenge were pre-selected to conduct a Cox proportional hazards analysis, which considers infection status and time to infection together with immune marker levels.”

  1. In line with the comment above, do the highest ADCC responses in Figure 3 correlate with the animals that exhibited protection against challenge? Based on the gross analysis, it doesn’t seem as though the ADCC responses bolster protection since the bivalent group #2 respond the best but have poor protection

We understand the reviewer’s line of reasoning. As described above, Figure 3 shows the overall immunogenicity of the vaccine regimens but does not consider protection status or time to infection, which are parameters in the Cox Models. It is not possible to truly provide a graphical representation of the hazard/risk that is calculated in a Cox Models for a continuous variable like the ADCC response. However, we were able to identify that ADCC levels against the challenge gp120 at the time of challenge were lower in animals that became infected earlier with fewer challenges, while this was not the case for other primary immune markers (Table 1).

  1. What is the amino acid diversity in the V1V2 loop region amongst the primary and booster Envs compared to the challenge Env? Does this explain the variability in protection phenotype across groups?

The reviewer raises a good point. Based on a secondary analysis, there is a good possibility that a match at HXB2 position 169 can explain at least some of the rank ordering of the protective effect observed (trivalent and pentavalent vaccines ranked better in terms of protection than the two bivalent vaccine regimens). The challenge SHIV Envelope, CH505, carries an R169 (Figure 2. Please refer to the PDF file for the figure). Both the trivalent and pentavalent vaccine include the CAP260 gp120, which also carries R169, and so in both cases one Env among the polyvalent mixture carries CH505 matched R169. In contrast, both bivalent vaccines carry only K169, the most common form.

Position 169 is of particular interest because a match in this position was the only V2 signature site for which a match to the vaccine was positively associated with vaccine efficacy in the RV144 trial (Rolland et al. PMID: 22960785). This is a highly variable position, and within the C clade it is most often K169 (67%), followed by R169 (8%), and a mixture of rarer amino acids makes up the other 25% (N169 is ~3%).

The trivalent and pentavalent vaccine carry mixtures of K169, R169, and N169; it is possible that diversity in this position in the vaccine cocktail may favor selection of antibodies that tolerate variation in position 169, and so not require a specific match. We have added this paragraph to the discussion to provide additional clarity to the manuscript (lines 319-331):

“Interestingly, when considering the V2 sequence of the vaccine regimens to the challenge Envelope, the Trivalent and Pentavalent include the immunogen CAP260.C gp120 which contains an Arginine at position 169 (R169) similar to the challenge Envelope CH505, while the other regimens included immunogens which all contained a Lysine (K169).  Position 169 is of particular interest because a match in this position was the only V2 signature site for which a match to the vaccine was positively associated with vaccine efficacy in the RV144 trial [12]. It Is also a highly variable position in the HIV Envelope: within clade C, K169 is present in 67% of Envelope sequences, followed by R169 (8%), and a mixture of rarer amino acids make up the remaining 25% (N169 is present in ~3% of Envelopes). The Trivalent and Pentavalent vaccine regimens carry mixtures of K169, R169, and N169; therefore, it is possible that diversity in this position in the vaccine cocktail favored selection of antibodies that tolerate variation in position 169 and so do not require a specific match.

  1. Table in Figure 1A is blurry/hard to read

We apologize for the resolution of the table in Figure 1A and have replaced it with a high-resolution table.

Round 2

Reviewer 2 Report

Comments and Suggestions for Authors

The authors have addressed all of my comments/concerns.